# Gaussian Process Prior Variational Autoencoders

**Francesco Paolo Casale**[†*]**, Adrian V Dalca**[‡§]**, Luca Saglietti**[†¶]**,
Jennifer Listgarten**[♯]**, Nicolo Fusi**[†]

[†] Microsoft Research New England, Cambridge (MA), USA
[‡] Computer Science and Artificial Intelligence Lab, MIT, Cambridge (MA), USA
[§] Martinos Center for Biomedical Imaging, MGH, HMS, Boston (MA), USA;
[¶] Italian Institute for Genomic Medicine, Torino, Italy
[♯] EECS Department, University of California, Berkeley (CA), USA.
[*] `frcasale@microsoft.com`

## Abstract

Variational autoencoders (VAE) are a powerful and widely-used class of models
to learn complex data distributions in an unsupervised fashion. One important
limitation of VAEs is the prior assumption that latent sample representations are in-
dependent and identically distributed. However, for many important datasets, such
as time-series of images, this assumption is too strong: accounting for covariances
between samples, such as those in time, can yield to a more appropriate model
specification and improve performance in downstream tasks. In this work, we
introduce a new model, the Gaussian Process (GP) Prior Variational Autoencoder
(GPPVAE), to specifically address this issue. The GPPVAE aims to combine the
power of VAEs with the ability to model correlations afforded by GP priors. To
achieve efficient inference in this new class of models, we leverage structure in
the covariance matrix, and introduce a new stochastic backpropagation strategy
that allows for computing stochastic gradients in a distributed and low-memory
fashion. We show that our method outperforms conditional VAEs (CVAEs) and an
adaptation of standard VAEs in two image data applications.

## 1 Introduction

Dimensionality reduction is a fundamental approach to compression of complex, large-scale data
sets, either for visualization or for pre-processing before application of supervised approaches.
Historically, dimensionality reduction has been framed in one of two modeling camps: the simple and
rich capacity language of neural networks; or the probabilistic formalism of generative models, which
enables Bayesian capacity control and provides uncertainty over latent encodings. Recently, these
two formulations have been combined through the Variational Autoencoder (VAE) (Kingma and
Welling, 2013), wherein the expressiveness of neural networks was used to model both the mean and
the variance of a simple likelihood. In these models, latent encodings are assumed to be identically
and independently distributed (*iid*) across both latent dimensions and samples. Despite this simple
prior, the model lacks conjugacy, exact inference is intractable and variational inference is used. In
fact, the main contribution of the Kingma *et al.* paper is to introduce an improved, general approach
for variational inference (also developed in Rezende et al. (2014)).

One important limitation of the VAE model is the prior assumption that latent representations of
samples are *iid*, whereas in many important problems, accounting for sample structure is crucial
for correct model specification and consequently, for optimal results. For example, in autonomous
driving, or medical imaging (Dalca et al., 2015; Lonsdale et al., 2013), high dimensional images are
correlated in time—an *iid* prior for these would not be sensible because, *a priori*, two images that
were taken closer in time should have more similar latent representations than images taken further
apart. More generally, one can have multiple sequences of images from different cars, or medical

image sequences from multiple patients. Therefore, the VAE prior should be able to capture multiple levels of correlations at once, including time, object identities, *etc.* A natural solution to this problem is to replace the VAE *iid* prior over the latent space with a Gaussian Process (GP) prior (Rasmussen, 2004), which enables the specification of sample correlations through a kernel function (Durrande et al., 2011; Gönen and Alpaydın, 2011; Wilson and Adams, 2013; Wilson et al., 2016; Rakitsch et al., 2013; Bonilla et al., 2007). GPs are often amenable to exact inference, and a large body of work in making computationally challenging GP-based models tractable can be leveraged (GPs naively scale cubically in the number of samples) (Gal et al., 2014; Bauer et al., 2016; Hensman et al., 2013; Csató and Opper, 2002; Quiñonero-Candela and Rasmussen, 2005; Titsias, 2009).

In this work, we introduce the Gaussian Process Prior Variational Autoencoder (GPPVAE), an extension of the VAE latent variable model where correlation between samples is modeled through a GP prior on the latent encodings. The introduction of the GP prior, however, introduces two main computational challenges. First, naive computations with the GP prior have cubic complexity in the number of samples, which is impractical in most applications. To mitigate this problem one can leverage several tactics commonly used in the GP literature, including the use of pseudo-inputs (Csató and Opper, 2002; Gal et al., 2014; Hensman et al., 2013; Quiñonero-Candela and Rasmussen, 2005; Titsias, 2009), Kronecker-factorized covariances (Casale et al., 2017; Stegle et al., 2011; Rakitsch et al., 2013), and low rank structures (Casale et al., 2015; Lawrence, 2005). Specifically, in the instantiations of GPPVAE considered in this paper, we focus on low-rank factorizations of the covariance matrix. A second challenge is that the *iid* assumption which guarantees unbiasedness of mini-batch gradient estimates (used to train standard VAEs) no longer holds due to the GP prior. Thus mini-batch GD is no longer applicable. However, for the applications we are interested in, comprising sequences of large-scale images, it is critical from a practical standpoint to avoid processing all samples simultaneously; we require a procedure that is both low in memory use and yields fast inference. Thus, we propose a new scheme for gradient descent that enables Monte Carlo gradient estimates in a distributable and memory-efficient fashion. This is achieved by exploiting the fact that sample correlations are only modeled in the latent (low-dimensional) space, whereas high-dimensional representations are independent when conditioning on the latent ones.

In the next sections we (i) discuss our model in the context of related work, (ii) formally develop the model and the associated inference procedure, (iii) compare GPPVAE with alternative models in empirical settings, demonstrating the advantages of our approach.

## 2 Related work

Our method is related to several extensions of the standard VAE that aim at improving the latent representation by leveraging auxiliary data, such as time annotations, pose information or lighting. An *ad hoc* attempt to induce structure on the latent space by grouping samples with specific properties in mini-batches was introduced in Kulkarni et al. (2015). More principled approaches proposed a semi-supervised model using a continuous-discrete mixture model that concatenates the input with auxiliary information (Kingma et al., 2014). Similarly, the conditional VAE (Sohn et al., 2015) incorporates auxiliary information in both the encoder and the decoder, and has been used successfully for sample generation with specific categorical attributes. Building on this approach, several models use the auxiliary information in an unconditional way (Suzuki et al., 2016; Pandey and Dukkipati, 2017; Vedantam et al., 2017; Wang et al., 2016; Wu and Goodman, 2018).

A separate body of related work aims at designing a more flexible variational posterior distributions, either by considering a dependence on auxiliary variables (Maaløe et al., 2016), by allowing structured encoder models (Siddharth et al., 2016), or by considering chains of invertible transformations that can produce arbitrarily complex posteriors (Kingma et al., 2016; Nalisnick et al., 2016; Rezende and Mohamed, 2015). In other work, a dependency between latent variables is induced by way of hierarchical structures at the level of the parameters of the variational family (Ranganath et al., 2016; Tran et al., 2015).

The extensions of VAEs most related to GPPVAE are those that move away from the assumption of an *iid* Gaussian prior on the latent representations to consider richer prior distributions (Jiang et al., 2016; Shu et al., 2016; Tomczak and Welling, 2017). These build on the observation that overly-simple priors can induce excessive regularization, limiting the success of such models (Chen et al., 2016; Hoffman and Johnson, 2016; Siddharth et al., 2017). For example, Johnson *et al.* proposed

composing latent graphical models with deep observational likelihoods. Within their framework, more flexible priors over latent encodings are designed based on conditional independence assumptions, and a conditional random field variational family is used to enable efficient inference by way of message-passing algorithms (Johnson et al., 2016).

In contrast to existing methods, we propose to model the relationship between the latent space and the auxiliary information using a GP prior, leaving the encoder and decoder as in a standard VAE (independent of the auxiliary information). Importantly, the proposed approach allows for modeling arbitrarily complex sample structure in the data. In this work, we specifically focus on disentangling sample correlations induced by different aspects of the data. Additionally, GPPVAE enables estimation of latent auxiliary information when such information is unobserved by leveraging previous work (Lawrence, 2005). Finally, using the encoder and decoder networks together with the GP predictive posterior, our model provides a natural framework for out-of-sample predictions of high-dimensional data, for virtually any configuration of the auxiliary data.

## 3 Gaussian Process Prior Variational Autoencoder

Assume we are given a set of samples (*e. g.,* images), each coupled with different types of auxiliary data (*e. g.,* time, lighting, pose, person identity). In this work, we focus on the case of two types of auxiliary data: *object* and *view* entities. Specifically, we consider datasets with images of *objects* in different *views*. For example, images of faces in different poses or images of hand-written digits at different rotation angles. In these problems, we know both which object (person or hand-written digit) is represented in each image in the dataset, and in which view (pose or rotation angle). Finally, each unique object and view is attached to a feature vector, which we refer to as an *object feature vector* and a *view feature vector*, respectively. In the face dataset example, object feature vectors might contain face features such as skin color or hair style, while view feature vectors may contain pose features such as polar and azimuthal angles with respect to a reference position. Importantly, as described and shown below, we can learn these feature vectors if not observed.

### 3.1 Formal description of the model

Let $N$ denote the number of samples, $P$ the number of unique objects and $Q$ the number of unique views. Additionally, let $\{\boldsymbol{y}_n\}_{n=1}^N$ denote $K$-dimensional representation for $N$ samples; let $\{\boldsymbol{x}_p\}_{p=1}^P$ denote $M$-dimensional object feature vectors for the $P$ objects; and let $\{\boldsymbol{w}_q\}_{q=1}^Q$ denote $R$-dimensional view feature vectors for the $Q$ views. Finally, let $\{\boldsymbol{z}_n\}_{n=1}^N$ denote the $L$-dimensional latent representations. We consider the following generative process for the observed samples (Fig 1a):

- the latent representation of object $p_n$ in view $q_n$ is generated from object feature vector $\boldsymbol{x}_{p_n}$ and view feature vector $\boldsymbol{w}_{q_n}$ as
$$\boldsymbol{z}_n = f(\boldsymbol{x}_{p_n}, \boldsymbol{w}_{q_n}) + \boldsymbol{\eta}_n, \text{ where } \boldsymbol{\eta}_n \sim \mathcal{N}\left(\mathbf{0}, \alpha \boldsymbol{I}_L\right); \tag{1}$$

- image $\boldsymbol{y}_n$ is generated from its latent representation $\boldsymbol{z}_n$ as
$$\boldsymbol{y}_n = g(\boldsymbol{z}_n) + \boldsymbol{\epsilon}_n, \text{ where } \boldsymbol{\epsilon}_n \sim \mathcal{N}\left(\mathbf{0}, \sigma_y^2 \boldsymbol{I}_K\right). \tag{2}$$

The function $f : \mathbb{R}^M \times \mathbb{R}^R \to \mathbb{R}^L$ defines how sample latent representations can be obtained in terms of object and view feature vectors, while $g : \mathbb{R}^L \to \mathbb{R}^K$ maps latent representations to the high-dimensional sample space.

We use a Gaussian process (GP) prior on $f$, which allows us to model sample covariances in the latent space as a function of object and view feature vectors. Herein, we use a convolutional neural network for $g$, which is a natural choice for image data (LeCun et al., 1995). The resulting marginal likelihood of the GPPVAE, is

$$p(\boldsymbol{Y} \mid \boldsymbol{X}, \boldsymbol{W}, \boldsymbol{\phi}, \sigma_y^2, \boldsymbol{\theta}, \alpha) = \int p(\boldsymbol{Y} \mid \boldsymbol{Z}, \boldsymbol{\phi}, \sigma_y^2) p(\boldsymbol{Z} \mid \boldsymbol{X}, \boldsymbol{W}, \boldsymbol{\theta}, \alpha) \, d\boldsymbol{Z}, \tag{3}$$

where $\boldsymbol{Y} = [\boldsymbol{y}_1, \dots, \boldsymbol{y}_N]^T \in \mathbb{R}^{N \times K}$, $\boldsymbol{Z} = [\boldsymbol{z}_1, \dots, \boldsymbol{z}_N]^T \in \mathbb{R}^{N \times L}$, $\boldsymbol{W} = [\boldsymbol{w}_1, \dots, \boldsymbol{w}_Q]^T \in \mathbb{R}^{Q \times R}$, $\boldsymbol{X} = [\boldsymbol{x}_1, \dots, \boldsymbol{x}_P]^T \in \mathbb{R}^{P \times M}$. Additionally, $\boldsymbol{\phi}$ denotes the parameters of $g$ and $\boldsymbol{\theta}$ the GP kernel parameters.

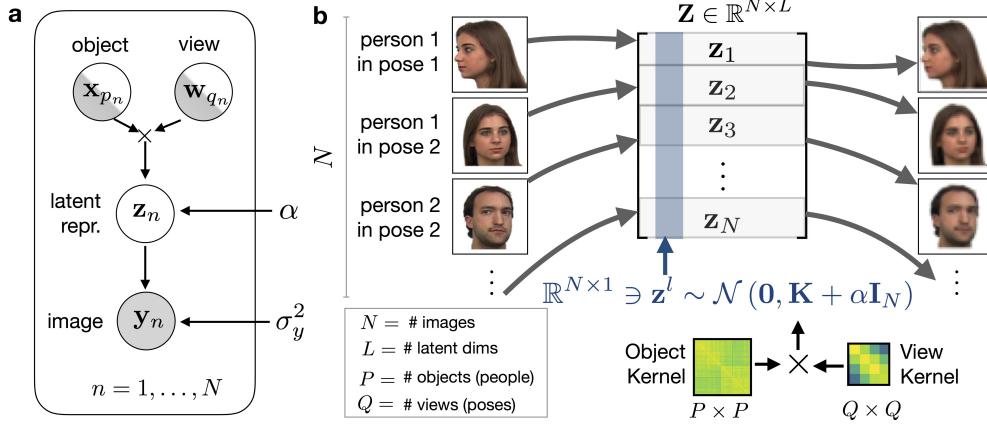

Figure 1: **(a)** Generative model underlying the proposed GPPVAE. **(b)** Pictorial representation of the inference procedure in GPPVAE. Each sample (here an image) is encoded in a low-dimensional space and then decoded to the original space. Covariances between samples are modeled through a GP prior on each column of the latent representation matrix $\boldsymbol{Z}$.

**Gaussian Process Model.** The GP prior defines the following multivariate normal distribution on latent representations:

$$p\left(\boldsymbol{Z} \mid \boldsymbol{X}, \boldsymbol{W}, \boldsymbol{\theta}, \alpha\right) = \prod_{l=1}^{L} \mathcal{N}\left(\boldsymbol{z}^l \mid \boldsymbol{0}, \boldsymbol{K_\theta}(\boldsymbol{X}, \boldsymbol{W}) + \alpha \boldsymbol{I}_N\right), \tag{4}$$

where $\boldsymbol{z}^l$ denotes the $l$-th column of $\boldsymbol{Z}$. In the setting considered in this paper, the covariance function $\boldsymbol{K_\theta}$ is composed of a view kernel that models covariances between views, and an object kernel that models covariances between objects. Specifically, the covariance between sample $n$ (with corresponding feature vectors $\boldsymbol{x}_{p_n}$ and $\boldsymbol{w}_{q_n}$) and sample $m$ (with corresponding feature vectors $\boldsymbol{x}_{p_m}$ and $\boldsymbol{w}_{q_m}$) is given by the factorized form (Bonilla et al., 2007; Rakitsch et al., 2013):

$$\boldsymbol{K_\theta}(\boldsymbol{X}, \boldsymbol{W})_{nm} = \mathcal{K}_{\boldsymbol{\theta}}^{(\text{view})}(\boldsymbol{w}_{q_n}, \boldsymbol{w}_{q_m}) \mathcal{K}_{\boldsymbol{\theta}}^{(\text{object})}(\boldsymbol{x}_{p_n}, \boldsymbol{x}_{p_m}). \tag{5}$$

**Observed versus unobserved feature vectors** Our model can be used when either one, or both of the view/sample feature vectors are unobserved. In this setting, we regard the unobserved features as latent variables and obtain a point estimate for them, similar to Gaussian process latent variable models (Lawrence, 2005). We have done so in our experiments.

## 3.2 Inference

As with a standard VAE, we make use of variational inference for our model. Specifically, we consider the following variational distribution over the latent variables

$$q_{\boldsymbol{\psi}}(\boldsymbol{Z} \mid \boldsymbol{Y}) = \prod_n \mathcal{N}\left(\boldsymbol{z}_n \mid \boldsymbol{\mu}_{\boldsymbol{\psi}}^z(\boldsymbol{y}_n), \text{diag}(\boldsymbol{\sigma}^{z\,2}_{\boldsymbol{\psi}}(\boldsymbol{y}_n))\right), \tag{6}$$

which approximates the true posterior on $\boldsymbol{Z}$. In Eq. (6), $\boldsymbol{\mu}_{\boldsymbol{\psi}}^z$ and $\boldsymbol{\sigma}_{\boldsymbol{\psi}}^z$ are the hyperparameters of the variational distribution and are neural network functions of the observed data, while $\boldsymbol{\psi}$ denotes the weights of such neural networks. We obtain the following evidence lower bound (ELBO):

$$\log p(\boldsymbol{Y} \mid \boldsymbol{X}, \boldsymbol{W}, \boldsymbol{\phi}, \sigma_y^2, \boldsymbol{\theta}) \geq \mathbb{E}_{\boldsymbol{Z} \sim q_{\boldsymbol{\psi}}}\left[\sum_n \log \mathcal{N}(\boldsymbol{y}_n \mid g_{\boldsymbol{\phi}}(\boldsymbol{z}_n), \sigma_y^2 \boldsymbol{I}_K) + \log p\left(\boldsymbol{Z} \mid \boldsymbol{X}, \boldsymbol{W}, \boldsymbol{\theta}, \alpha\right)\right] +$$

$$+ \frac{1}{2} \sum_{nl} \log(\boldsymbol{\sigma}^{z\,2}_{\boldsymbol{\psi}}(\boldsymbol{y}_n)_l) + \text{const.} \tag{7}$$

**Stochastic backpropagation.** We use stochastic backpropagation to maximize the ELBO (Kingma and Welling, 2013; Rezende et al., 2014). Specifically, we approximate the expectation by sampling from a reparameterized variational posterior over the latent representations, obtaining the following loss function:

$$l\left(\phi, \psi, \theta, \alpha, \sigma_y^2\right) =$$

$$= NK\log \sigma_y^2 + \underbrace{\sum_n \frac{\left\|\boldsymbol{y}_n - g_\phi(\boldsymbol{z}_{\psi_n})\right\|^2}{2\sigma_y^2}}_{\text{reconstruction term}} - \underbrace{\log p\left(\boldsymbol{Z}_\psi \mid \boldsymbol{X}, \boldsymbol{W}, \boldsymbol{\theta}, \alpha\right)}_{\text{latent-space GP term}} + \underbrace{\frac{1}{2}\sum_{nl}\log(\boldsymbol{\sigma}^{z\,2}_\psi(\boldsymbol{y}_n)_l)}_{\text{regularization term}}, \quad (8)$$

which we optimize with respect to $\phi, \psi, \theta, \alpha, \sigma_y^2$. Latent representations $\boldsymbol{Z}_\psi = \left[\boldsymbol{z}_{\psi_1}, \ldots, \boldsymbol{z}_{\psi_N}\right] \in \mathbb{R}^{N \times L}$ are sampled using the re-parameterization trick Kingma and Welling (2013),

$$\boldsymbol{z}_{\psi_n} = \mu_\psi^z(\boldsymbol{y}_n) + \boldsymbol{\epsilon}_n \odot \sigma^z_\psi(\boldsymbol{y}_n), \ \boldsymbol{\epsilon}_n \sim \mathcal{N}(\boldsymbol{0}, \boldsymbol{I}_{L \times L}), \ n = 1, \ldots, N, \quad (9)$$

where $\odot$ denotes the Hadamard product. Full details on the derivation of the loss can be found in Supplementary Information.

**Efficient GP computations.** Naive computations in Gaussian processes scale cubically with the number of samples (Rasmussen, 2004). In this work, we achieve linear computations in the number of samples by assuming that the overall GP kernel is low-rank. In order to meet this assumption, we (i) exploit that in our setting the number of views, $Q$, is much lower than the number of samples, $N$, and (ii) impose a low-rank form for the object kernel ($M \ll N$). Briefly, as a result of these assumptions, the total covariance is the sum of a low-rank matrix and the identity matrix $\boldsymbol{K} = \boldsymbol{V}\boldsymbol{V}^T + \alpha\boldsymbol{I}$, where $\boldsymbol{V} \in \mathbb{R}^{N \times H}$ and $H \ll N$ [1]. For this covariance, computation of the inverse and the log determinant, which have cubic complexity for general covariances, can be recast to have complexity, $O(NH^2 + H^3 + HNK)$ and $O(NH^2 + H^3)$, respectively, using the Woodbury identity (Henderson and Searle, 1981) and the determinant lemma (Harville, 1997):

$$\boldsymbol{K}^{-1}\boldsymbol{M} = \frac{1}{\alpha}\boldsymbol{I} - \frac{1}{\alpha}\boldsymbol{V}(\alpha\boldsymbol{I} + \boldsymbol{V}^T\boldsymbol{V})^{-1}\boldsymbol{V}^T\boldsymbol{M}, \quad (10)$$

$$\log|\boldsymbol{K}| = NL\log\alpha + \log\left|\boldsymbol{I} + \frac{1}{\alpha}\boldsymbol{V}^T\boldsymbol{V}\right|, \quad (11)$$

where $\boldsymbol{M} \in \mathbb{R}^{N \times K}$. Note a low-rank approximation of an arbitrary kernel can be obtained through the fully independent training conditional approximation (Snelson and Ghahramani, 2006), which makes the proposed inference scheme applicable in a general setting.

**Low-memory stochastic backpropagation.** Owing to the coupling between samples from the GP prior, mini-batch gradient descent is no longer applicable. However, a naive implementation of full gradient descent is impractical as it requires loading the entire dataset into memory, which is infeasible with most image datasets. To overcome this limitation, we propose a new strategy to compute gradients on the whole dataset in a low-memory fashion. We do so by computing the first-order Taylor series expansion of the GP term of the loss with respect to both the latent encodings and the prior parameters, at each step of gradient descent. In doing so, we are able to use the following procedure:

1. Compute latent encodings from the high-dimensional data using the encoder. This step can be performed in data mini-batches, thereby imposing only low-memory requirements.

2. Compute the coefficients of the GP-term Taylor series expansion using the latent encodings. Although this step involves computations across all samples, these have low-memory requirements as they only involve the low-dimensional representations.

3. Compute a proxy loss by replacing the GP term by its first-order Taylor series expansion, which locally has the same gradient as the original loss. Since the Taylor series expansion is linear in the latent representations, gradients can be easily accumulated across data mini-batches, making this step also memory-efficient.

4. Update the parameters using these accumulated gradients.

Full details on this procedure are given in Supplementary Information.

## 3.3 Predictive posterior

We derive an approximate predictive posterior for GPPVAE that enables out-of-sample predictions of high-dimensional samples. Specifically, given training samples $\boldsymbol{Y}$, object feature vectors $\boldsymbol{X}$, and view feature vectors $\boldsymbol{W}$, the predictive posterior for image representation $\boldsymbol{y}_\star$ of object $p_\star$ in view $q_\star$ is given by

$$p(\boldsymbol{y}_\star \,|\, \boldsymbol{x}_\star, \boldsymbol{w}_\star, \boldsymbol{Y}, \boldsymbol{X}, \boldsymbol{W}) \approx \int \underbrace{p(\boldsymbol{y}_\star \,|\, \boldsymbol{z}_\star)}_{\text{decode GP prediction}} \underbrace{p(\boldsymbol{z}_\star \,|\, \boldsymbol{x}_\star, \boldsymbol{w}_\star, \boldsymbol{Z}, \boldsymbol{X}, \boldsymbol{W})}_{\text{latent-space GP predictive posterior}} \underbrace{q(\boldsymbol{Z} \,|\, \boldsymbol{Y})}_{\text{encode training data}} \; d\boldsymbol{z}_\star d\boldsymbol{Z} \quad (12)$$

where $\boldsymbol{x}_\star$ and $\boldsymbol{w}_\star$ are object and feature vectors of object $p_\star$ and view $q_\star$ respectively, and we dropped the dependency on parameters for notational compactness. The approximation in Eq. (12) is obtained by replacing the exact posterior on $\boldsymbol{Z}$ with the variational distribution $q(\boldsymbol{Z} \,|\, \boldsymbol{Y})$ (see Supplementary Information for full details). From Eq. (12), the mean of the GPPVAE predictive posterior can be obtained by the following procedure: (i) encode training image data in the latent space through the encoder, (ii) predict latent representation $\boldsymbol{z}_\star$ of image $\boldsymbol{y}_\star$ using the GP predictive posterior, and (iii) decode latent representation $\boldsymbol{z}_\star$ to the high-dimensional image space through the decoder.

# 4 Experiments

We focus on the task of making predictions of unseen images, given specified auxiliary information. Specifically, we want to predict the image representation of object $p$ in view $q$ when that object was never observed in that view, but assuming that object $p$ was observed in at least one other view, and that view had been observed for at least one other object. For example, we may want to predict the pose of a person appearing in our training data set without having seen that person in that pose. To do so, we need to have observed that pose for other people. This prediction task gets at the heart of what we want our model to achieve, and therefore serves as a good evaluation metric.

## 4.1 Methods considered

In addition to the GPPVAE presented (GPPVAE-joint), we also considered a version with a simpler optimization scheme (GPPVAE-dis). We also considered two extensions of the VAE that can be used for the task at hand. Specifically, we considered:

- **GPPVAE with joint optimization (GPPVAE-joint)**, where autoencoder and GP parameters were optimized jointly. We found that convergence was improved by first training the encoder and the decoder through standard VAE, then optimizing the GP parameters with fixed encoder and decoder for 100 epochs, and finally, optimizing all parameters jointly. Out-of-sample predictions from GPPVAE-joint were obtained by using the predictive posterior in Eq. (12);

- **GPPVAE with disjoint optimization (GPPVAE-dis)**, where we first learned the encoder and decoder parameters through standard VAE, and then optimized the GP parameters with fixed encoder and decoder. Again, out-of-sample predictions were obtained by using the predictive posterior in Eq. (12);

- **Conditional VAE (CVAE)** (Sohn et al., 2015), where view auxiliary information was provided as input to both the encoder and decoder networks (Figure S1, S2). After training, we considered the following procedure to generate an image of object $p$ in view $q$. First, we computed latent representations of all the images of object $p$ across all the views in the training data (in this setting, CVAE latent representations are supposedly independent from the view). Second, we averaged all the obtained latent representations to obtain a unique representation of object $p$. Finally, we fed the latent representation of object $p$ together with out-of-sample view $q$ to the CVAE decoder. As an alternative implementation, we also tried to consider the latent representation of a random image of object $p$ instead of averaging, but the performance was worse–these results are not included;

- **Linear Interpolation in VAE latent space (LIVAE)**, which uses linear interpolation between observed views of an object in the latent space learned through standard VAE in order to predict unobserved views of the same object. Specifically, denoting $z_1$ and $z_2$ as the latent representations of images of a given object in views $r_1$ and $r_2$, a prediction for the image of that same object in an intermediate view, $r_\star$, is obtained by first linearly interpolating between $z_1$ and $z_2$, and then projecting the interpolated latent representation to the high-dimensional image space.

Consistent with the L2 reconstruction error appearing in the loss off all the aforementioned VAEs (*e. g.,* Eq. (18)), we considered pixel-wise mean squared error (MSE) as the evaluation metric. We used the same architecture for encoder and decoder neural networks in all methods compared (see Figure S1, S2 in Supplementary Information). The architecture and $\sigma_y^2$, were chosen to minimize the ELBO loss for the standard VAE on a validation set (Figure S3, Supplementary Information). For CVAE and LIVAE, we also considered the alternative strategy of selecting the value of $\sigma_y^2$ that maximizes out-of-sample prediction performance on the validation set (the results for these two methods are in Figure S4 and S5). All models were trained using the Adam optimizer (Kingma and Ba, 2014) with standard parameters and a learning rate of $0.001$. When optimizing GP parameters with fixed encoder and decoder, we observed that higher learning rates led to faster convergence without any loss in performance, and thus we used a higher learning rate of $0.01$ in this setting.

## 4.2 Rotated MNIST

**Setup.** We considered a variation of the MNIST dataset, consisting of rotated images of handwritten "3" digits with different rotation angles. In this setup, objects correspond to different draws of the digit "3" while views correspond to different rotation states. View features are observed scalars, corresponding to the attached rotation angles. Conversely, object feature vectors are unobserved and learned from data—no draw-specific features are available.

**Dataset generation.** We generated a dataset from 400 handwritten versions of the digit three by rotating through $Q = 16$ evenly separated rotation angles in $[0, 2\pi)$, for a total of $N = 6,400$ samples. We then kept $90\%$ of the data for training and test, and the rest for validation. From the training and test sets, we then randomly removed $25\%$ of the images to consider the scenario of incomplete data. Finally, the set that we used for out-of-sample predictions (test set) was created by removing one of the views (*i. e.,* rotation angles) from the remaining images. This procedure resulted in $4,050$ training images spanning 15 rotation angles and 270 test images spanning one rotation angle.

**Autoencoder and GP model.** We set the dimension of the latent space to $L = 16$. For encoder and decoder neural networks we considered the convolutional architecture in Figure S1. As view kernel, we considered a periodic squared exponential kernel taking rotation angles as inputs. As object kernel, we considered a linear kernel taking the object feature vectors as inputs. As object feature vectors are unobserved, we learned from data—their dimensionality was set to $M = 8$. The resulting composite kernel $\boldsymbol{K}$, expresses the covariance between images $n$ and $m$ in terms of the corresponding rotations angles $w_{q_n}$ and $w_{q_m}$ and object feature vectors $\boldsymbol{x}_{p_n}$ and $\boldsymbol{x}_{p_m}$ as

$$\boldsymbol{K_\theta}(\boldsymbol{X}, \boldsymbol{w})_{nm} = \beta \underbrace{\exp\left(-\frac{2\sin^2|w_{q_n} - w_{q_m}|}{\nu^2}\right)}_{\text{rotation kernel}} \cdot \underbrace{\boldsymbol{x}_{p_n}^T \boldsymbol{x}_{p_m}}_{\text{digit draw kernel}}, \tag{13}$$

where $\beta \geq 0$ and $\nu \geq 0$ are kernel hyper-parameters learned during training of the model (Rasmussen, 2004), and we set $\boldsymbol{\theta} = \{\beta, \nu\}$.

**Results.** GPPVAE-joint and GPPVAE-dis yielded lower MSE than CVAE and LIVAE in the interpolation task, with GPPVAE-joint performing significantly better than GPPVAE-dis ($0.0280 \pm 0.0008$ for GPVAE-joint vs $0.0306 \pm 0.0009$ for GPPVAE-dis, $p < 0.02$, **Fig. 2a,b**). Importantly, GPPVAE-joint learns different variational parameters than a standard VAE (Fig. 2c,d), used also by GPPVAE-dis, consistent with the fact that GPPVAE-joint performs better by adapting the VAE latent space using guidance from the prior.

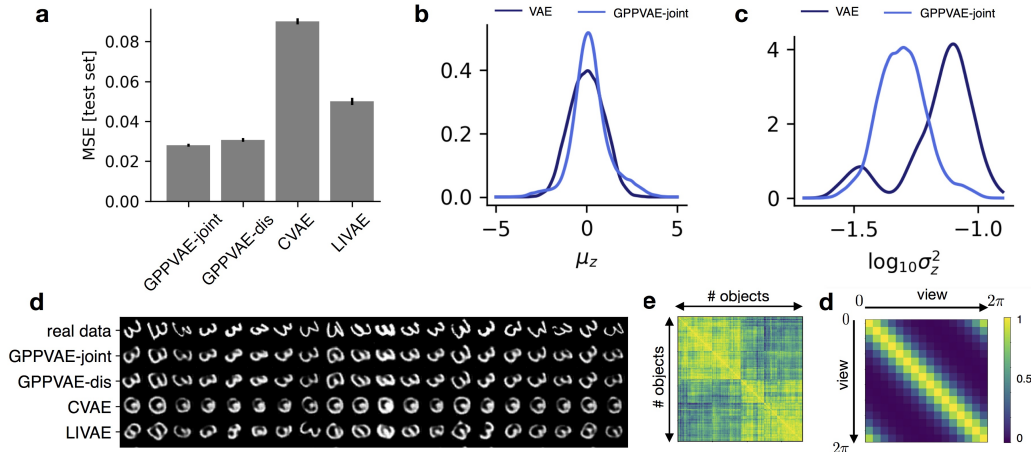

Figure 2: Results from experiments on rotated MNIST. (**a**) Mean squared error on test set. Error bars represent standard error of per-sample MSE. (**b**) Empirical density of estimated means of $q_\psi$, aggregated over all latent dimensions. (**c**) Empirical density of estimated log variances of $q_\psi$. (**d**) Out-of-sample predictions for ten random draws of digit "3" at the out-of-sample rotation state. (**e**, **f**) Object and view covariances learned through GPPVAE-joint.

### 4.3   Face dataset

**Setup.**   As second application, we considered the Face-Place Database (3.0) (Righi et al., 2012), which contains images of people faces in different poses. In this setting, objects correspond to the person identities while views correspond to different poses. Both view and object feature vectors are unobserved and learned from data. The task is to predict images of people face in orientations that remained unobserved.

**Data.**   We considered 4,835 images from the Face-Place Database (3.0), which includes images of faces for 542 people shown across nine different poses (frontal and 90, 60, 45, 30 degrees left and right[2]). We randomly selected $80\%$ of the data for training ($n = 3,868$), $10\%$ for validation ($n = 484$) and $10\%$ for testing ($n = 483$). All images were rescaled to $128 \times 128$.

**Autoencoder and GP model.**   We set the dimension of the latent space to $L = 256$. For encoder and decoder neural networks we considered the convolutional architecture in Figure S2. We consider a full-rank covariance as a view covariance (only nine poses are present in the dataset) and a linear covariance for the object covariance ($M = 64$).

**Results.**   GPPVAE-jointand GPPVAE-disyielded lower MSE than CVAE and LIVAE (**Fig. 2a**,**b**). In contrast to the MNIST problem, the difference between GPPVAE-joint andGPPVAE-dis was not significant ($0.0281 \pm 0.0008$ for GPPVAE-joint vs $0.0298 \pm 0.0008$ for GPPVAE-dis). Importantly, GPPVAE-joint was able to dissect people (object) and pose (view) covariances by learning people and pose kernel jointly (**Fig. 2a**,**b**).

## 5   Discussion

We introduced GPPVAE, a generative model that incorporates a GP prior over the latent space. We also presented a low-memory and computatationally efficient inference strategy for this model, which makes the model applicable to large high-dimensional datasets. GPPVAE outperforms natural baselines (CVAE and linear interpolations in the VAE latent space) when predicting out-of-sample test images of objects in specified views (*e. g.,* pose of a face, rotation of a digit). Possible future

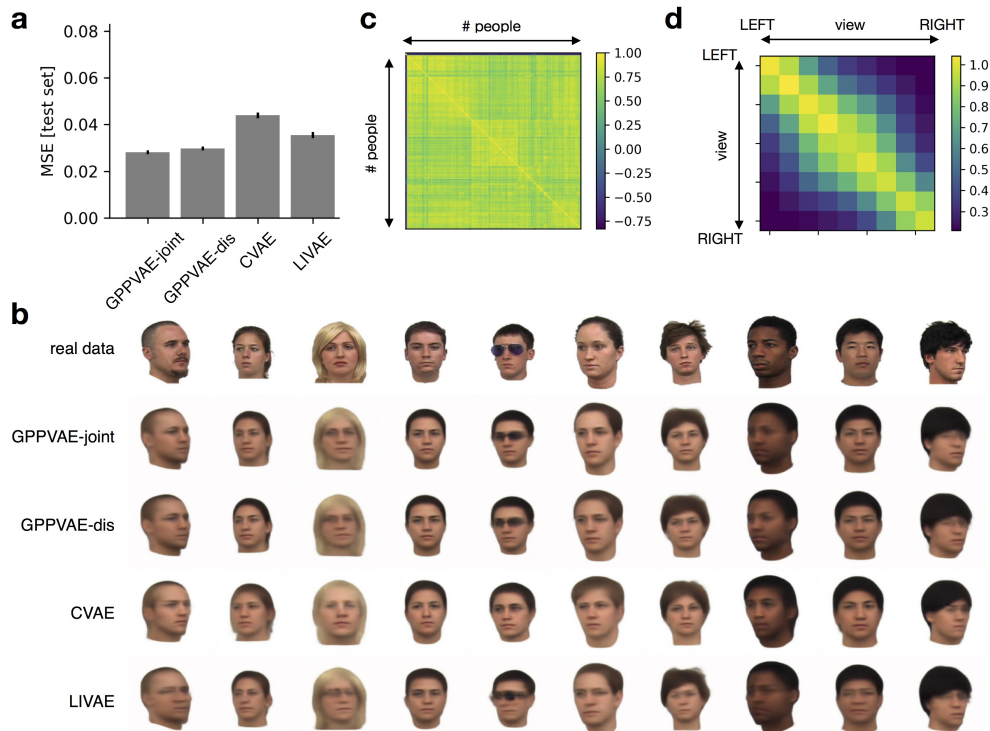

Figure 3: Results from experiments on the face dataset. (**a**) Mean squared error on test set (**b**) Out-of-sample predictions of people faces in out-of-sample poses. (**c**, **d**) Object and view covariances learned through GPPVAE-joint.

work includes augmenting the GPPVAE loss with a discriminator function, similar in spirit to a GAN (Goodfellow et al., 2014), or changing the loss to be perception-aware (Hou et al., 2017) (see results from preliminary experiments in Figure S6). Another extension is to consider approximations of the GP likelihood that fully factorize over data points (Hensman et al., 2013); this could further improve the scalability of our method.

**Code availability**

An implementation of GPPVAE is available at `https://github.com/fpcasale/GPPVAE`.

**Acknowledgments**

Stimulus images courtesy of Michael J. Tarr, Center for the Neural Basis of Cognition and Department of Psychology, Carnegie Mellon University, http:// www.tarrlab.org. Funding provided by NSF award 0339122.

## Footnotes

[1] For example, if both the view and the object kernels are linear, we have $\boldsymbol{V} = [\boldsymbol{X}_{:,1} \odot \boldsymbol{W}_{:,1}, \boldsymbol{X}_{:,1} \odot \boldsymbol{W}_{:,2}, \ldots, \boldsymbol{X}_{:,M} \odot \boldsymbol{W}_{:,Q}] \in \mathbb{R}^{N \times H}$.

[2] We could have used the pose angles as view feature scalar similar to the application in rotated MNIST, but purposely ignored these features to consider a more challenging setting were neither object and view features are observed.

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
