[Supplementary Material]

# Supplementary Information for Gaussian Process Prior Variational Autoencoders

## Contents

## 1 Supplementary Methods

### 1.1 ELBO derivation

We assume the following posterior distribution on latent variables

$$q_{\boldsymbol{\psi}}(\boldsymbol{Z} \,|\, \boldsymbol{Y}) = \prod_n \mathcal{N}\left( \boldsymbol{z}_n \,|\, \boldsymbol{\mu}_{\boldsymbol{\psi}}^z(\boldsymbol{y}_n), \mathrm{diag}(\boldsymbol{\sigma}^{z\,2}_{\boldsymbol{\psi}}(\boldsymbol{y}_n)) \right), \tag{1}$$

where $\boldsymbol{\psi}$ are variational parameters. The evidence lower bound (ELBO) can be derived as follows:

$$
\begin{aligned}
\log p(\boldsymbol{Y} \,|\, \boldsymbol{X}, \boldsymbol{W}, \boldsymbol{\phi}, \sigma_y^2, \boldsymbol{\theta}) &= \log \int \frac{p(\boldsymbol{Y} \,|\, \boldsymbol{z}, \boldsymbol{\phi}, \sigma_y^2) p(\boldsymbol{Z} \,|\, \boldsymbol{X}, \boldsymbol{W}, \boldsymbol{\theta}, \alpha)}{q_{\boldsymbol{\psi}}(\boldsymbol{Z} \,|\, \boldsymbol{Y})} q_{\boldsymbol{\psi}}(\boldsymbol{Z} \,|\, \boldsymbol{Y}) d\boldsymbol{Z} \\
&\geq \int \log \left( \frac{p(\boldsymbol{Y} \,|\, \boldsymbol{z}, \boldsymbol{\phi}, \sigma_y^2) p(\boldsymbol{Z} \,|\, \boldsymbol{X}, \boldsymbol{W}, \boldsymbol{\theta}, \alpha)}{q_{\boldsymbol{\psi}}(\boldsymbol{Z} \,|\, \boldsymbol{Y})} \right) q_{\boldsymbol{\psi}}(\boldsymbol{Z} \,|\, \boldsymbol{Y}) d\boldsymbol{Z} \\
&= \mathbb{E}_{\boldsymbol{Z} \sim q_{\boldsymbol{\psi}}} \left[ \sum_n \log \mathcal{N}(\boldsymbol{y}_n \,|\, g_{\boldsymbol{\phi}}(\boldsymbol{z}_n), \sigma_y^2 \boldsymbol{I}_K) + \log p(\boldsymbol{Z} \,|\, \boldsymbol{X}, \boldsymbol{W}, \boldsymbol{\theta}, \alpha) \right] - \int \log q_{\boldsymbol{\psi}}(\boldsymbol{Z} \,|\, \boldsymbol{Y}) q_{\boldsymbol{\psi}}(\boldsymbol{Z} \,|\, \boldsymbol{Y}) d\boldsymbol{Z} \\
&= \mathbb{E}_{\boldsymbol{Z} \sim q_{\boldsymbol{\psi}}} \left[ \sum_n \log \mathcal{N}(\boldsymbol{y}_n \,|\, g_{\boldsymbol{\phi}}(\boldsymbol{z}_n), \sigma_y^2 \boldsymbol{I}_K) + \log p(\boldsymbol{Z} \,|\, \boldsymbol{X}, \boldsymbol{W}, \boldsymbol{\theta}, \alpha) \right] + \frac{1}{2} \sum_{nl} \log(\boldsymbol{\sigma}^{z\,2}_{\boldsymbol{\psi}}(\boldsymbol{y}_n)_l) + \mathrm{const}
\end{aligned}
$$

### 1.2 Loss derivation

First, we approximate the expectation by sampling. Specifically, we sample a latent representation $\boldsymbol{Z}_{\boldsymbol{\psi}} = \left[ \boldsymbol{z}_{\boldsymbol{\psi}\,1}, \ldots, \boldsymbol{z}_{\boldsymbol{\psi}\,N} \right] \in \mathbb{R}^{N \times L}$ from the posterior as:

$$\boldsymbol{z}_{\boldsymbol{\psi}\,n} = \boldsymbol{\mu}_{\boldsymbol{\psi}}^z(\boldsymbol{y}_n) + \boldsymbol{\epsilon}_n \odot \boldsymbol{\sigma}^z_{\boldsymbol{\psi}}(\boldsymbol{y}_n), \ \boldsymbol{\epsilon}_n \sim \mathcal{N}(\boldsymbol{0}, \boldsymbol{I}_{L \times L}), \ n = 1, \ldots, N, \tag{2}$$

where we used the reparametrization trick to separate the noisy generation of samples from the model parameters. The resulting approximate ELBO is

$$\text{ELBO} \quad \approx \quad \sum_n \log \mathcal{N}(\boldsymbol{y}_n \,|\, g_{\boldsymbol{\phi}}(\boldsymbol{z}_{\boldsymbol{\psi}_n}), \sigma_y^2 \boldsymbol{I}_K) + \log p\left(\boldsymbol{Z}_{\boldsymbol{\psi}} \,|\, \boldsymbol{X}, \boldsymbol{W}, \boldsymbol{\theta}, \alpha\right) + \frac{1}{2} \sum_{nl} \log(\boldsymbol{\sigma^z}^2_{\boldsymbol{\psi}}(\boldsymbol{y}_n)_l) + \text{const}$$

Finally, as we optimize $\sigma_y^2$ on a validation set (and not on the training set in order to avoid overfitting), maximization of the approximate ELBO on the training set is equivalent to minimizing the following cost:

$$\mathcal{L}\left(\boldsymbol{\phi}, \boldsymbol{\psi}, \boldsymbol{\theta}, \alpha, \sigma_y^2\right) = \frac{1}{K} \underbrace{\sum_i \left(\boldsymbol{y}_i - g_{\boldsymbol{\phi}}(\boldsymbol{\omega}_{\boldsymbol{\psi}_n})\right)^2}_{\text{reconstruction term}} - \frac{\lambda}{L} \left[ \underbrace{\log p\left(\boldsymbol{Z}_{\boldsymbol{\psi}} \,|\, \boldsymbol{X}, \boldsymbol{W}, \boldsymbol{\theta}, \alpha\right)}_{\text{latent-space GP model}} + \underbrace{\frac{1}{2} \sum_{nl} \log(\boldsymbol{\sigma^z}^2_{\boldsymbol{\psi}}(\boldsymbol{y}_n)_l)}_{\text{regularization}} \right], \quad (3)$$

where $K$ is the number of pixels in the image and $\lambda$ is a trade-off parameter balancing data reconstruction and latent space prior regularization.

**Selection of $\lambda$.** We select the value of $\lambda$ based on a standard VAE and use the same value for all compared models. Specifically, we optimize the VAE loss for different values of $\lambda$ and select the value for which the VAE ELBO is maximal on a validation set. In order to compute the VAE validation ELBO we need to estimate optimal value of $\sigma_y^2$ for every value of $\lambda$ in the grid. Given a certain value of $\lambda = \hat{\lambda}$, the optimal value of $\sigma_y^2$ can be estimated as

$$\sigma_y^{2\,(\text{val})} = \frac{1}{N^{(\text{val})}} \sum_{n=1}^{N^{(\text{val})}} \left(\boldsymbol{y}_n^{(\text{val})} - g_{\boldsymbol{\phi}_{\hat{\lambda}}}(\boldsymbol{z}_{\boldsymbol{\psi}_{\hat{\lambda}}}{}_n^{(\text{val})})\right)^2, \quad (4)$$

where $N^{(\text{val})}$ is the number of samples in the validation set and $(\boldsymbol{\phi}_{\hat{\lambda}}, \boldsymbol{\psi}_{\hat{\lambda}})$ are the values of the encoder/decoder parameters after training for $\lambda = \hat{\lambda}$.

## 1.3 Fast low-rank computations

Parameter inference in Gaussian models scales cubically with the number of observations. To preform fast parameter inference in our Gaussian model, we use the fact that the total covariance is the sum of low-rank matrix and the identity matrix:

$$\boldsymbol{K} = \boldsymbol{V}\boldsymbol{V}^T + \alpha \boldsymbol{I} \quad (5)$$

where $\boldsymbol{V} \in \mathbb{R}^{N \times O}$ and $O \ll N$. Using the Woodbury identity [1] and the determinant lemma [2], the linear system $\boldsymbol{K}^{-1}\boldsymbol{M}$ with $M \in \mathbb{R}^{N \times K}$ and the log determinant of $\boldsymbol{K}$ can be computed as:

$$\boldsymbol{K}^{-1}\boldsymbol{M} = \frac{1}{\alpha}\boldsymbol{I} - \frac{1}{\alpha}\boldsymbol{V}(\alpha\boldsymbol{I} + \boldsymbol{V}^T\boldsymbol{V})^{-1}\boldsymbol{V}^T\boldsymbol{M}, \quad (6)$$

$$\text{logdet}\boldsymbol{K} = NL\text{logdet}\alpha + \text{logdet}(\boldsymbol{I} + \frac{1}{\alpha}\boldsymbol{V}^T\boldsymbol{V}), \quad (7)$$

which have $O(NO^2 + O^3 + ONK)$ and $O(NO^2 + O^3)$ complexities, respectively.

## 1.4 Implementation of low-memory stochastic backprobagation

For a single latent dimension the GP prior introduces the following term in the ELBO:

$$\log p\left(\boldsymbol{z}_{\boldsymbol{\psi}} \,|\, \boldsymbol{X}, \boldsymbol{W}, \boldsymbol{\theta}, \alpha\right) = -\frac{1}{2}\boldsymbol{z}_{\boldsymbol{\psi}}^T \boldsymbol{K}_\theta(\boldsymbol{X}, \boldsymbol{V})^{-1}\boldsymbol{z}_{\boldsymbol{\psi}} - \frac{1}{2}\text{logdet}\boldsymbol{K}_\theta(\boldsymbol{X}, \boldsymbol{V}) \quad (8)$$

In the equation above, $z_\psi$ functionally depends on the image space representations and thus a naive computation of full gradient descent would require loading the entire high-dimensional dataset in memory, which would be unfeasible form many high-dimensional image datasets. To overcome this limitation, we recast computations in a form where full-matrix operations only take place in the low dimensional space, while the dependency on the nested derivatives, which involve high memory loads, is linearized and mini-batch operations are therefore allowed. This can be achieved by considering a first-order Taylor expansion of the GP prior term. Specfically, using that $\boldsymbol{K_\theta} = \boldsymbol{V_\theta V_\theta^T} + \alpha \boldsymbol{I}$, and collecting all parameters in $\boldsymbol{\xi} = \{\boldsymbol{\psi}, \boldsymbol{\theta}, \alpha\}$, we can rewrite the the term in Eq. (8) in this functional form $f(\boldsymbol{z}(\boldsymbol{\xi}), \boldsymbol{V}(\boldsymbol{\xi}), \alpha(\boldsymbol{\xi}))$. The first-order Taylor expansion of $f(\boldsymbol{z}(\boldsymbol{\xi}), \boldsymbol{V}(\boldsymbol{\xi}), \alpha(\boldsymbol{\xi}))$ around $(\boldsymbol{z}_{\boldsymbol{\xi}_0}, \boldsymbol{V}_{\boldsymbol{\xi}_0}, \alpha_{\boldsymbol{\xi}_0})$ is:

$$f(\boldsymbol{z}(\boldsymbol{\xi}), \boldsymbol{V}(\boldsymbol{\xi}), \alpha(\boldsymbol{\xi})) \approx \boldsymbol{a}^T \boldsymbol{z}(\boldsymbol{\xi}) + \mathrm{tr}\left(\boldsymbol{B}^T \boldsymbol{V}(\boldsymbol{\xi})\right) + c\alpha(\boldsymbol{\xi}) + \mathrm{const.} \tag{9}$$

where

$$\boldsymbol{a} = \left(\frac{\partial f}{\partial \boldsymbol{z}}\right)_{\boldsymbol{\xi}_0} = \left(\boldsymbol{K}^{-1}\boldsymbol{z}\right)_{\boldsymbol{\xi}_0} \tag{10}$$

$$\boldsymbol{B} = \left(\frac{\partial f}{\partial \boldsymbol{V}}\right)_0 = \left(-\boldsymbol{K}^{-1}\boldsymbol{z}\boldsymbol{z}^T K^{-1}\boldsymbol{V} + \boldsymbol{K}^{-1}\boldsymbol{V}\right)_{\boldsymbol{\xi}_0} \tag{11}$$

$$c = \left(\frac{\partial f}{\partial \alpha}\right)_{\boldsymbol{\xi}_0} = \frac{1}{2}\left(-\boldsymbol{z}^T\boldsymbol{K}^{-1}\boldsymbol{K}^{-1}\boldsymbol{z} + \mathrm{tr}(\boldsymbol{K}^{-1})\right)_{\boldsymbol{\xi}_0} \tag{12}$$

Note that this approximation, applied at every step of gradient descent, locally preserves the gradients. Thus we can engineer the following low-memory four-step procedure for full gradient descent:

- produce and store latent noise realizations ($\boldsymbol{\epsilon}$) used for the reparametrization trick;

- obtain latent variable representations $\boldsymbol{z}$, combining the outputs of the encoder and the noise $\boldsymbol{\epsilon}$. We employ mini-batch forward propagation for this step;

- evaluate $\boldsymbol{a}$, $\boldsymbol{B}$ and $c$ across all samples. Note that this step has low-memory requirements as it only involes low-dimensional representations;

- exploit the local Taylor expansion as a proxy for our optimization. As this function is linear in the data, we can accumulate its gradient in a mini-batch fashion, by passing mini-batches of $\boldsymbol{\epsilon}$, $\boldsymbol{Y}$, $\boldsymbol{a}$ and $\boldsymbol{B}$;

- update the parameters $\boldsymbol{\xi}$ using the full gradients as in standard gradient descent.

Finally, we note that in our specific setting $\boldsymbol{a}$, $\boldsymbol{B}$ and $c$ can be computed linearly on the number of sample because of the low rank structure of $\boldsymbol{K}$ (see previous section).

## 1.5 Out of sample predictions.

We here derive an approximate predictive posterior for GPPVAE. Specifically, given training images $\boldsymbol{Y}$, object representations $\boldsymbol{X}$ and view representations $\boldsymbol{V}$ the predictive posterior for image $\boldsymbol{y}_\star$ representing

object $\boldsymbol{x}_\star$ in view $\boldsymbol{v}_\star$ is

$$
\begin{aligned}
p(\boldsymbol{y}_\star \,|\, \boldsymbol{x}_\star, \boldsymbol{v}_\star, \boldsymbol{Y}, \boldsymbol{X}, \boldsymbol{V}) &= \frac{p(\boldsymbol{y}_\star, \boldsymbol{Y} \,|\, \boldsymbol{x}_\star, \boldsymbol{v}_\star, \boldsymbol{X}, \boldsymbol{V})}{p(\boldsymbol{Y} \,|\, \boldsymbol{X}, \boldsymbol{V})} && (13) \\[2mm]
&= \frac{1}{p(\boldsymbol{Y} \,|\, \boldsymbol{X}, \boldsymbol{V})} \int p(\boldsymbol{y}_\star \,|\, \boldsymbol{z}_\star) p(\boldsymbol{Y} \,|\, \boldsymbol{Z}) \underbrace{p(\boldsymbol{z}_\star, \boldsymbol{Z} \,|\, \boldsymbol{x}_\star, \boldsymbol{v}_\star, \boldsymbol{X}, \boldsymbol{V})}_{\text{joint distribution}} d\boldsymbol{z}_\star d\boldsymbol{Z} && (14) \\[2mm]
&= \frac{1}{p(\boldsymbol{Y} \,|\, \boldsymbol{X}, \boldsymbol{V})} \int p(\boldsymbol{y}_\star \,|\, \boldsymbol{z}_\star) p(\boldsymbol{Y} \,|\, \boldsymbol{Z}) \underbrace{p(\boldsymbol{z}_\star \,|\, \boldsymbol{x}_\star, \boldsymbol{v}_\star, \boldsymbol{Z}, \boldsymbol{X}, \boldsymbol{V})}_{\text{predictive posterior}} \underbrace{p(\boldsymbol{Z} \,|\, \boldsymbol{X}, \boldsymbol{V})}_{\text{GP prior on training}} d\boldsymbol{z}_\star d\boldsymbol{Z} && (15) \\[2mm]
&= \int p(\boldsymbol{y}_\star \,|\, \boldsymbol{z}_\star) \underbrace{p(\boldsymbol{z}_\star \,|\, \boldsymbol{x}_\star, \boldsymbol{v}_\star, \boldsymbol{Z}, \boldsymbol{X}, \boldsymbol{V})}_{\text{predictive posterior}} \underbrace{\frac{p(\boldsymbol{Y} \,|\, \boldsymbol{Z}) p(\boldsymbol{Z} \,|\, \boldsymbol{X}, \boldsymbol{V})}{p(\boldsymbol{Y} \,|\, \boldsymbol{X}, \boldsymbol{V})}}_{\text{posterior on } \boldsymbol{Z}} d\boldsymbol{z}_\star d\boldsymbol{Z} && (16) \\[2mm]
&\approx \int p(\boldsymbol{y}_\star \,|\, \boldsymbol{z}_\star) \underbrace{p(\boldsymbol{z}_\star \,|\, \boldsymbol{x}_\star, \boldsymbol{v}_\star, \boldsymbol{Z}, \boldsymbol{X}, \boldsymbol{V})}_{\text{predictive posterior}} \underbrace{q(\boldsymbol{Z} \,|\, \boldsymbol{Y})}_{\text{approx. post. on } \boldsymbol{Z}} d\boldsymbol{z}_\star d\boldsymbol{Z} && (17)
\end{aligned}
$$

where we dropped dependencies from model parameters to simplify the notation.

# 2 Supplementary Figures

**Figure S1  Neural network architecture used in the MNIST experiments.** (a) Encoder archi­tecture; (b) Decoder architecture. The same encoder/decoder architectures are used for both VAE and GPPVAE. For CVAE, we still use the same architecture but we provide view representations (rotation angles) as inputs to both the encoder and the decoder. For the encoder, we provide rotation angles to the first layer (by adding additional channels to the input image) and again before the dense layer (by stacking them to the vector produced by the last convolution). Similarly, for the decoder, we provide rotation angles as inputs to the first layer (by stacking them with the 16-dimensional latent variable) and again before the first convolution layer (by adding them as additional channels to the 8x8x4 tensor fed to the first convolution layer). For a fair comparison with GPPVAE, we account for periodicity in the view representation by providing both the angle sine and cosine, i.e. $\boldsymbol{w} = [\sin\phi, \cos\phi]$, where $\phi$ is the rotation angle.

**Figure S2 Neural network architecture used in the face dataset experiments.** (a) Encoder architecture; (b) Decoder architecture. The same encoder/decoder architectures are used for both VAE and GPPVAE. For CVAE, we still use the same architecture but we provide view information as one hot encoding of the 9 different poses to both the encoder and the decoder networks. For the encoder, we provide view representations to the first layer (by adding additional channels to the input image) and again before the dense layer (by stacking them to the vector produced by the last convolution). Similarly, for the decoder, we provide view representations as inputs to the first layer (by stacking them with the 16-dimensional latent variable) and again before the first convolution layer (by adding them as additional channels to the 8x8x4 tensor fed to the first convolution layer).

**Figure S3 Selection of the trade-off parameter between fit-to-data and latent-space model.** The trade-off parameter was selected as to maximize VAE ELBO. We here show the VAE validation lower bound as a function of the trade-off parameter for the MNIST (a) and the face dataset (b).

**Figure S4** For CVAE and LIVAE, we also considered the alternative strategy of selecting the value of the trade-off parameter $\lambda$ that maximizes prediction performance on the validation set. We refer to these additional methods as CVAE-opt and LIVAE-opt. We here show the selection of $\lambda$ for CVAE-opt (a) and LIVAE-opt (b) and their performance (c-d) for the experiments in MNIST.

**Figure S5** Analogous to Figure S4 but for the face data experiments.

**Figure S6** Out-of-sample predictions obtained from GPPVAE-joint when considering the perceptual loss introduced in [3].