[Reviews · NeurIPS 2018]

Reviewer 1



Summary of the Paper: This paper describes an extension of the variational autoencoder to take into account correlations among the latent variables z that are used to generate each observed samples. These correlations are modeled through a Gaussian process prior in which the input variables are additional variables related to object or view information associated to each data point. The method is compared with other approaches from the literature that can potentially address these correlations on two datasets. Strengths: - Well written paper. - Complete related work section. - Nice and illustrative experiments. Weaknesses: - The optimization of the proposed method is complicated and not very well described in the paper. - I have missed a baseline to compare with which does not take into account correlations. Quality: The quality of the paper is high. Originality: As far as I know this work is original. Significance: The results obtained indicate that the proposed approach can obtain better results than the methods the authors compare with. Therefore, I believe that the results are significant. Other comments: It is not described in the paper how to optimize and quickly evaluate the lower bound on the log probability of the data. In particular, the introduction of correlations among the z's variables implies that the lower bound is no longer expressed as a sum across the data points. This makes the optimization process more complicated. Furthermore, GPs do not scale well with the number of instances. The authors should provide more details about this has been addressed in the main manuscript. Not in the supplementary material. I have missed some comparison with a simple baseline to show that these correlations are indeed important. For example, a standard VAE.

Reviewer 2



In this manuscript the authors propose a method for introducing auxiliary information regarding object 'view' (i.e., information ancillary to the class 'label') into the method of (Bayesian) variational autoencoders for object classification. The challenge here, of course, is not simply writing down an additional layer in the hierarchical structure but in devising an efficient computational implementation, which the authors do achieve (via a low rank GP approximation, Taylor series expansion and sensible memory management). Overall I find this to be a strong contribution, although I would accept that with respect to the guidelines for authors this could be argued to be more of an 'incremental' extension of an exisiting method rather than a new paradigm in itself. A minor concern I have regards the presentation of the method: it seems to me that the strength and applicability of this method is more in terms of allowing the introduction of auxiliary information from well-understood transformations of the base objects in a dataset with relatively complete transformation instances for most objects; rather than for more general regularisation such as for potentially heterogeneous datasets including the autonomous driving and perhaps even the medical imaging applications suggested in the introduction. The reason I say this is because choosing an appropriate kernel over the auxiliary data + object class space requires a degree of 'supervised' understanding of the problem and controlling/learning the low rank representation will be challenging when the data 'design' is sparse. Likewise, for the range of applications for which a separable kernel is appropriate. (Related: I do not understand the kernel description in eqn 9: is the x^T x term correct? I was expecting something more like a distance than a vector length.) After author feedback: Yes, I understand now to see that these are feature vectors and hence taking the inner product in the feature space for the kernel. Thanks

Reviewer 3



This paper presents a method for improving Variational Auto-encoder performance when additional information (such a view angle or image time) is available. The solution proposed is simply to place a GP prior over the latent representations. The main technical difficulty the authors faced was in reducing the computational complexity of the GP and in particular allowing batch stochastic gradient descent methods to be applied, which is complicated by the non-parametric nature of the GP. I wonder if the authors considered using a sparse GP with the fully independent conditional (FIC) assumption [1]. Using this approach, when we conditioning on the inducing variables, the likelihood factorises over data points and we can apply batch SGD directly without any problems. I thought failure to consider this approach was a weak part of the paper. I thought the experiments were adequate and show the method does 1) improve performance and 2) out perform the benchmark algorithm. I think we would be very disappointed if the first of these had not been true however. Overall I thought this was an OK paper. I don’t think the paper provides any great insights, although the method of pre-training the VAE then learning the GP parameters, then joint training was appreciated. I would also have liked to know how well the system worked when used in GPLVM mode, with unobserved feature vectors. However the general approach is reasonable and the exposition is fine. Typos etc. Line 40 word missing: A natural solution to this problem is [to?] replace the … Line 194 doesn’t quite make sense: …we found that convergence was improved by first doing inference only the VAE Line 204 word missing: …where we [omit?] some parameters for notational compactness. [1] Quinonero-Candela, Rasmussen, A Unifying View of Sparse Approximate Gaussian Process Regression, JMRL 2005.